# On the Acquisition of Differential Object Marking in Child Heritage Spanish: Bilingual Education, Exposure, and Age Effects (In Memory of Phoebe Search)

Patrick D. Thane 

College of Education, University of Massachusetts Amherst, Amherst, MA 01003, USA; pthane@umass.edu

**Abstract:** Studies on school-aged children have been infrequent in research on Spanish as a heritage language. The present study explored how dual-language immersion education, patterns of heritage language use, proficiency, and age shape child Spanish heritage speakers' production and selection of differential object marking (DOM). A total of 57 English–Spanish bilingual children and 18 Spanish-dominant adults completed sentence completion and morphology selection tasks. Results revealed that the group of heritage speaker children that produced and selected the differential object marker most frequently was the seventh and eighth grade children (ages 12–14, the oldest in the study) who had completed a dual-language immersion program. Different factors accounted for variability in each task: bilingual education and proficiency affected the production of DOM, while age affected selection. Heritage speakers selected DOM more frequently than they produced this structure. These findings have implications for theories of heritage language acquisition that emphasizes that language experience and exposure account for differences between heritage speakers and argue for the dissociation of production from underlying syntactic knowledge. The data also argue that heritage speakers may possess a bilingual alignment for DOM, whereby underlying receptive knowledge is modulated by cumulative exposure, while production depends more on bilingual education and proficiency in Spanish.

**Keywords:** heritage languages; differential object marking; dual-language education

## 1. Introduction

Research on heritage language (HL) acquisition has been extensive, but only recently has this field begun to experience a "paradigm shift" (Giancaspro et al. 2022, p. 484) that moves away from strictly comparing heritage speakers (HSs) to other groups of speakers (both multilingual and monolingual) and towards an understanding of what contributes to differences between HSs. As Rothman et al. (2023) point out, the spectrum of possible outcomes of HL acquisition is wide, and HSs often converge on the norms of speakers dominant in the same language. As a result, it is essential to understand which factors contribute to the emergence of these differences between HSs. As exemplified by articles within this special issue (e.g., López Otero 2023; Martínez Vera et al. 2023; Solano-Escobar and Cuza 2023), there are multiple experiential variables that can account for these differences and that enrich our understanding of the many possible paths of acquisition of Spanish as a HL.

Traditional approaches to HL acquisition have claimed that HSs' grammars diverge from baselines who are dominant in the same language (e.g., Montrul 2008, 2013). While this approach is often accurate in predicting that HSs innovate beyond the language in their input, Putnam and Sánchez (2013) propose a framework that provides specific predictions concerning the process of HL acquisition and maintenance, which is particularly useful for studying within-group differences and for accounting for variability between and within individual speakers. Crucially, evaluating the role of multiple experiential variables provides a more holistic account of the acquisition of HLs more generally.

Putnam and Sánchez's (2013) approach builds on the Feature Reassembly Hypothesis (e.g., Lardiere 2009) from second language acquisition, which argues that the acquisition of functional features[1] requires the appropriate mapping of syntactic and semantic information onto inflectional morphology. Such an account dissociates morphological difficulties that may surface in production from underlying syntactic competence. In adapting Lardiere's (2009) framework to contexts of HL acquisition, Putnam and Sánchez (2013) and Perez-Cortes et al. (2019) argue that HSs remap the features of the more dominant language onto their HL, first in production and subsequently at the representational level. This exemplifies bilingual alignments (Sánchez 2019), in which HSs may possess strong receptive knowledge that converges on the syntactic representations of speakers dominant in the same language, even if they produce structures that show crosslinguistic influence from another language. To summarize, Putnam and Sánchez (2013) predict that HSs reassemble the features of their HL based upon crosslinguistic influence from the more dominant language, first in production and subsequently at the underlying level, and that the degree of maintenance is contingent upon sustained processing of and experiences with the HL.

Despite its multiple testable predictions, this framework has not been the topic of extensive scholarship with bilingual children. The present study complements this growing body of research that has primarily concentrated on adults with a focus on Spanish–English bilingual children. HL acquisition research on school-aged children has been scant, leading Montrul (2018, p. 534) to argue that this population is the "missing link" in this line of study. Understanding the trajectory of HL development across childhood allows us to distinguish between theories that posit a protracted yet sustained path of acquisition across these years and those that argue that HSs attrite their grammatical knowledge by adulthood due to decreases in exposure. For example, it is implicit in Putnam and Sánchez's (2013) feature reassembly account of HL acquisition and maintenance that HSs acquire functional features and subsequently reassemble them due to crosslinguistic influence on bundles of lexical items. Testing children of different ages, as well as those who differ along multiple dimensions that may affect HL acquisition, allows us to trace patterns of development and the path of acquisition that have often remained elusive in HL research.

New to this line of work is the exploration of how education shapes the acquisition of Spanish in school-aged children. Previous research on German-dominant HSs of French, Italian, and Turkish has shown that literacy and educational experience have an important impact on HL acquisition (Bayram et al. 2017; Kupisch et al. 2014; Kupisch and Rothman 2018). In these studies, speakers who attended school or who reported higher levels of literacy in their HL were more likely to pattern with speakers dominant in the same language in their command of multiple morphosyntactic properties.

Kupisch and Rothman (2018) argue that bilingual education purportedly provides HS children with greater quantity (more exposure to the HL) and quality (greater lexical diversity through content-specific vocabulary and greater syntactic diversity) of input when compared to monolingual education in the socially dominant language, which has been tied to HL attrition (Anderson 2001; Merino 1983; Silva-Corvalán 2014). Therefore, literacy and bilingual education comprise additional variables that can characterize differences in outcomes of HL acquisition between individual HSs. In fact, Putnam and Sánchez (2013, p. 490) reference bilingual education as a possible variable that leads to the acquisition and conservation of HLs.

To this effect, Bayram et al. (2017, p. 935) propose convincingly that "increased literacy seems to convey . . . some protection against CLI [crosslinguistic influence]". This may also be the case in the preliteracy period: in a longitudinal study, Barnett et al. (2007) showed that within six months, Spanish-dominant children as young as age three began to experience HL vocabulary loss while enrolled in an English-only preschool. In contrast, HSs enrolled in DLI programs in which half of academic and preliteracy instruction took place in Spanish showed vocabulary growth, unlike their peers.

Consequently, studying children with different levels of HL use and literacy using productive and receptive tasks is an essential next step within Putnam and Sánchez's (2013)

framework. Potowski (2007, p. 164) elucidates the importance of such a study: "What about . . . a study that compares L1 heritage speakers in a dual immersion program with their counterparts in . . . programs taught mostly in English?" To address this opportunity for research, this study compared English-dominant Spanish HS children and adolescents enrolled in an English–Spanish DLI school with those in an English-only school in their production and receptive knowledge of differential object marking (DOM). DOM is a syntactic structure that previous experimental research has revealed to be highly variable in child and adult HSs of Spanish (e.g., Arechabaleta Regulez and Montrul 2023; Cuza et al. 2019; Guijarro-Fuentes et al. 2017; Guijarro-Fuentes and Marinis 2011; Hur 2020; Jegerski and Sekerina 2020; Montrul 2004; Montrul and Bowles 2009; Montrul et al. 2015; Montrul and Sánchez-Walker 2013; Sagarra et al. 2019). In so doing, this study makes four key contributions to HL acquisition research. Firstly, it provides a better understanding of how individual differences in overall HL frequency of use and proficiency affect bilingual children and adolescents. Secondly, it explores the role of bilingual education in acquiring Spanish as a HL. Thirdly, it tests the predictions concerning differences between productive and receptive knowledge (e.g., Perez-Cortes et al. 2019) with bilingual children, and finally, it evaluates two age groups to plot the developmental trajectory that HSs follow during this understudied age range.

The following section provides an overview of DOM, as well as the previous research on its acquisition by monolinguals and HS children and adults. Subsequently, the research questions and hypotheses, as well as participants and methods, are presented, followed by an analysis of results and individual differences. This article concludes with a discussion of findings and how they relate to the existing bodies of research on the roles of language experience and exposure in the acquisition of Spanish as a HL.

## 2. Differential Object Marking in Spanish

Spanish is one of approximately 300 known languages that feature DOM, through which some objects receive overt case marking based on semantic characteristics (Bossong 1991). In Spanish, DOM involves the use of the dative marker *a* to mark some direct objects, particularly those that are animate and specific (Torrego 1998; Zagona 2002).[2] Dative objects are also marked categorically with *a* regardless of animacy or specificity. Since multiple studies have found variation in the use of DOM in monolingual varieties (Callen and Miller 2021; Reina et al. 2021; Requena 2022), and it is optional in certain contexts such as with animals and in some relative clauses (see Sagarra et al. 2019), the present study evaluates HSs' knowledge of this grammatical structure with proper nouns referring to a specific person (e.g., *Juanito*). These instances of DOM are maximally animate and specific and do not show variability in monolingual populations, and it is precisely this context that Aissen (2003) claims is the most prototypical occurrence of DOM crosslinguistically along a scale of animacy.

Torrego (1998) claims that DOM is a morphological realization of inherent case in Spanish in which animate and specific direct objects move overtly from within the VP to *spec,vP* to check an interpretable D-feature. While English and Spanish have structural case, only Spanish has an inherent case system that is realized through DOM. DOM has important implications for meaning, as shown in the contrast between sentences (1) and (2). Specifically, sentence (1) has a non-canonical VS word order, where the subject of the verb *ver* ('to see') is *Juana*, while in (2), there is a null subject of the same verb whose object is *Juana*, as indicated by the dative marker *a*. In such cases, DOM disambiguates between the subject and direct object of the verb when both are animate. This facilitates the freer word order that Spanish exhibits when compared to English, the dominant language of most Spanish HSs in the United States.

| (1) | | Ve | Juana. | |
|---|---|---|---|---|
| | | See-3PS | Juana. | |
| | | | | Juana sees. |
| (2) | | Ve | a | Juana. |
| | | Ø See-3PS | DOM | Juana. |
| | | | | She (null subject) sees Juana. |

For Spanish-speaking children, the acquisition of DOM requires the semantic and pragmatic ability to recognize which referents are animate and specific, as well as the syntactic ability to distinguish between subjects and objects and to mark inherent case. Working within the feature reassembly approach to second language acquisition, Guijarro-Fuentes (2012) argues that the development of DOM requires mapping these syntactic and semantic entailments onto the dative marker *a* and using them in the appropriate contexts (e.g., with animate and specific direct objects, but not others). Within Putnam and Sánchez's (2013) feature-oriented approach to HL acquisition, HSs such as those in the present study, particularly those with lower levels of exposure to Spanish, may obviate the need for the dative marker *a* due to crosslinguistic influence from English, which does not have a DOM system. Specifically, this framework would predict that low exposure would result in the reassembly of the interpretable D-feature responsible for the raising of the direct object from VP-internal position to *spec,vP*, first in production and subsequently at the underlying representational level. The consequence could be that HSs do not need to mark animate and specific direct objects (or do so optionally). Conversely, following these predictions, HSs such as those who attend a DLI school and who purportedly have a higher quantity of exposure to Spanish should show stronger knowledge of DOM in production and at the receptive level.

DOM is a fruitful area for research on the acquisition of Spanish as a HL because, while it has important implications for meaning and word order, it has low perceptual salience, which has been tied to its acquisition (e.g., Montrul et al. 2015; Sagarra et al. 2019). Therefore, children may require extensive input in Spanish to overhear and subsequently acquire DOM. Furthermore, bilingual children's knowledge of DOM continues to be variable into the preschool period (Ticio 2015), as discussed in the following section. Since recent research has shown that HSs begin to experience a shift in dominance towards English concurrent with the onset of schooling (Barnett et al. 2007; Castilla-Earls et al. 2019; Hiebert and Rojas 2021) and around when DOM is acquired, sustained exposure to the HL through bilingual education during this period may be particularly crucial for the acquisition of this structure.

### 2.1. Acquisition of DOM by Spanish-Speaking Children

In a longitudinal study of four Spanish-speaking children, Rodríguez-Mondoñedo (2008) found that participants produced DOM with 98% accuracy by the age of three. However, one of the children was bilingual in Spanish and Catalan and exhibited the greatest omission of DOM. In a longitudinal corpus analysis, Ticio (2015) reported that seven English–Spanish bilingual children showed considerable rates of optionality between ages 3 and 6. For bilinguals, DOM production rates were as low as 25% in the expected contexts, while four age-matched monolingual children produced the *a* marker at a rate of 70%; however, following Requena (2022), many instances of DOM omission in this dataset referred to contexts that have been shown to exhibit dialectal variation.

Optional production of DOM has also been found in experimental research with school-aged bilingual children. Cuza et al. (2019) carried out an experimental study with 15 English–Spanish bilingual children between the ages of 6–7 and 11–12. These researchers found that HSs differed from age-matched monolingual peers and Spanish-dominant bilingual adults in their rates of DOM production; the latter group produced this structure categorically. While monolingual children's DOM production increased with age, there was no similar effect in the HSs' data. The researchers claimed that their findings point towards incomplete acquisition of DOM by bilingual children, which invites the question of how and why individual HSs differed from one another.

Data from older children come from a set of studies by Guijarro-Fuentes and Marinis (2011) and Guijarro-Fuentes et al. (2017), who explored the production and acceptability judgments of English–Spanish bilinguals and Spanish monolinguals between ten and fifteen years of age. These studies evaluated the use of DOM in multiple semantic contexts, testing both productive and receptive knowledge. Results showed that monolingual children produced more DOM than their bilingual peers, particularly with animate and specific direct objects. However, HSs patterned similarly to monolingual peers in their acceptability judgments. Monolinguals also showed variability in their DOM knowledge, which highlights that sustained exposure to the HL during the time period tested may be particularly important for the acquisition of this structure by children with less input than monolinguals. Patterns of current exposure and age did not modulate results, although proficiency did account for individual variability. The asymmetries between production and receptive knowledge and the role of proficiency, which has been taken as a proxy for HL exposure (e.g., López Otero and Jimenez 2022; Giancaspro and Sánchez 2021), support Putnam and Sánchez's (2013) predictions.

*2.2. Differential Object Marking in Adult Spanish Heritage Speakers*

Research with adult Spanish HSs has also found considerable optionality in the use of DOM using multiple methodologies. HSs show great variability ranging from full omission to categorical production of DOM (Montrul and Sánchez-Walker 2013), but higher-proficiency speakers tend to pattern similarly to monolingual or Spanish-dominant comparison groups (Arechabaleta Regulez and Montrul 2023; Hur 2020; Montrul 2004; Montrul and Bowles 2009). Two other studies have found that HSs' patterns of HL use also modulate their production of DOM. Firstly, Montrul and Sánchez-Walker (2013) reported that current exposure to Spanish, but not age of acquisition of English, accounted for variance in the production of DOM for both children and adults. Secondly, Hur (2020) demonstrated that participants' ratings of word frequency accounted for HSs' variable production of DOM, arguing that bilinguals' exposure to individual lexical items explained variability. Since each of the variables reviewed here represents HSs' patterns of current exposure to the HL,[3] there is evidence across studies that HL experience modulates the extent to which individual speakers command DOM in Spanish.

In contrast to the clear evidence regarding the role of exposure, the effect of age remains unclear. On one hand, Montrul and Sánchez-Walker (2013) found that simultaneous and sequential adult HSs produced DOM across an oral narrative and controlled production task more than children. However, the authors did not report whether there were age effects within the child group, making it difficult to determine at which point children converge on the adult-like DOM system. On the other hand, Cuza et al. (2019), Guijarro-Fuentes and Marinis (2011), and Guijarro-Fuentes et al. (2017) did not find age effects for their participants, some of whom were within the age range tested here. Most recently, Thane (forthcoming) replicated these findings, whereby there was a steady increase in DOM production from fifth grade (ages 10–11) to adulthood, but the differences between age groups of children did not achieve statistical significance. This aligns with each of the previous studies evaluating age by showing a slow yet steady increase over time. As stated previously, age effects are useful for understanding the path of HL acquisition or restructuring that is generally only examined with adults or preschool children, which presents an opportunity for further study.

## 3. The Study

The present study addresses the areas of research reviewed throughout the previous sections by exploring English-dominant HS children's acquisition of DOM with proper nouns that are animate and specific. The use of DOM in these contexts appears to be invariable, which minimizes the potential confound between HSs' innovations and general variation that is also observable in monolingual communities (e.g., Callen 2023; Guijarro-Fuentes et al. 2017; Requena 2022). Tapping productive and receptive knowledge as well

as individual HSs' patterns of use and proficiency is well-positioned to contribute novel evidence from school-aged children to Putnam and Sánchez's (2013) framework. Finally, exploring age effects and the role of bilingual education informs how these variables can affect the course of HL development. Although this study does incorporate comparisons with Spanish-dominant adults, its primary focus contributes to the "paradigm shift" (Giancaspro et al. 2022, p. 484) in HL acquisition research by exploring how age, productive versus receptive knowledge, bilingual education, and patterns of exposure (as measured through frequency of use and proficiency) affect variation between HSs. To do so, four research questions were proposed:

1.  Do Spanish HS children who attend a DLI school show differences in their production and selection of DOM with animate and specific direct objects when compared (A) to Spanish-dominant bilingual adults and (B) to age-matched HSs in monolingual English schools?

    Cuza et al. (2019) showed that Spanish-dominant adults used DOM categorically in the contexts tested. Therefore, it was predicted that these participants would produce and select DOM categorically in this study as well. Since previous research has found that adult HSs with high proficiency in and frequent exposure to the HL converge on the production rates of Spanish-dominant adults (e.g., Montrul and Bowles 2009; Montrul and Sánchez-Walker 2013), those participants in this study who attended a DLI school would plausibly also produce and select DOM at ceiling like Spanish-dominant adults, given their more frequent exposure to Spanish. However, given that Putnam and Sánchez (2013) predict that HSs with less exposure and activation, such as those who do not receive bilingual education, will reassemble the features of their HL, it was predicted that HSs in a monolingual English school would produce and select DOM less frequently than age-matched peers in DLI due to the relationship between exposure and greater crosslinguistic influence from English.

2.  Do frequency of HL use and morphosyntactic proficiency affect individual variability in DOM production and selection with animate and specific direct objects?

    Previous studies have found that frequency of use of Spanish and morphosyntactic proficiency modulate individual participants' knowledge of DOM (Guijarro-Fuentes et al. 2017; Montrul 2004; Montrul and Bowles 2009; Montrul and Sánchez-Walker 2013). Therefore, the same variables were predicted to account for variability in both the production and selection of DOM with animate and specific direct objects across tasks. Such a finding would argue that current levels of exposure to Spanish, as measured through frequency of use and proficiency, account for HSs' knowledge of DOM.

3.  Do Spanish HS children show increased command of DOM in animate and specific contexts with age?

    As stated above, age effects are difficult to interpret across previous studies. Although evidence is not clear, a logical hypothesis based on the available evidence comparing children and adults (Montrul and Sánchez-Walker 2013; Thane forthcoming) is that command of this structure increases during the late school period. Following this study, it was proposed that children in the seventh and eighth grades would show increased production and selection of DOM compared to participants in the fifth grade.

4.  Do Spanish HS children select DOM with animate and specific direct objects more frequently than they produce it?

    Guijarro-Fuentes and Marinis (2011) and Guijarro-Fuentes et al. (2017) provide evidence that bilingual children were more variable in their DOM production than in their acceptability judgments. This aligns with Putnam and Sánchez's (2013) predictions that HSs exhibit asymmetries between production and underlying syntactic

knowledge as measured on a receptive task. Therefore, it was predictable that HSs in the present study would select DOM on a receptive task more frequently than they would produce it.

*3.1. Participants*

A total of 57 HS children and 18 Spanish-dominant bilingual adults (SDBAs) participated in this study. All children attended either a DLI or a monolingual English school and were part of one of two age groups (fifth grade, ages 10–11, or seventh/eighth grade, ages 12–14). These children were predominantly second-generation sequential bilinguals who spoke Spanish with their parents. Most children were from Mexican-American families, although some were exposed to other Caribbean or Central American varieties.[4] There were also participants whose parents were from other Spanish-speaking regions. The HSs formed four participant groups: fifth graders in the English–Spanish bilingual school (BES-5), fifth graders in the monolingual English school (ME-5), seventh/eighth graders in the bilingual English–Spanish school (BES-7/8), and seventh/eighth graders in the monolingual English school (ME-7/8). These age groups were incorporated because they represent the end stages of primary and middle school DLI programs. Table 1 represents the division of children by school and grade groupings. Although this four-way division of the HSs who participated in this study was used in the descriptive statistics, type of education (DLI versus monolingual English) and age group (fifth grade versus seventh/eighth grade) were incorporated as separate variables in the multivariate analyses.

**Table 1.** Division of participants by grade and school.

| School | 5th Grade | 7th/8th Grade | Total by School |
|---|---|---|---|
| Bilingual English–Spanish | 19 | 13 | **32** |
| Monolingual English (ME) | 14 | 11 | **25** |
| **Total by group** | **33** | **24** | **57** |

The children who attended the DLI school received 50% of their academic instruction in Spanish from kindergarten through fifth grade and then continued to attend intensive Spanish as a heritage language class during the middle school years (therefore, the BES-7/8 group had fewer hours of exposure to Spanish at the time of the experiment than the BES-5 group). In order to participate in the experiment, children needed to have attended the DLI school for at least half of the primary years (since second grade).[5] In contrast, the children in the monolingual English school had not received any portion of their academic instruction in Spanish. It is important to note that while the present project evaluated Spanish language use across multiple contexts, it was not recorded whether individual participants attended extracurricular HL programs. Nevertheless, the two schools were matched (within 1%) for socioeconomic background, number of Latinx families, and number of English language learners. At least two thirds of students' parents reported Spanish as the dominant language at home on school demographic reports, and almost all participants in this study indicated that they spoke predominantly Spanish at home, making them sequential bilinguals. Each group's frequency of use of Spanish across six contexts (maximum 30 points), morphosyntactic proficiency in Spanish (maximum 18 points), and number of monolingual Spanish-speaking parents are summarized in Table 2.

Finally, the SDBA group was included because these participants represent the adult-like system of DOM that comprises the input that HSs receive in Spanish. Since Montrul and Sánchez-Walker (2013) show that bilingual adults experienced attrition of DOM, it was essential to incorporate a group of adults that represents the input that the HS children in this study receive, but who are also bilingual (see Pascual y Cabo and Rothman 2012 for an argument in support of this methodology). All SDBAs had received at least their primary and middle school education in one of seven Spanish-speaking countries. Their average length of residence in the United States was 9.5 years, and their proficiency in English

ranged from intermediate to superior. They had retained high morphosyntactic proficiency in Spanish (average 47.7/50) as measured using an adult proficiency test used extensively in previous research (e.g., Duffield and White 1999; Montrul and Slabakova 2003). This group represented the input that HSs receive from caregivers in diverse bilingual communities with speakers from multiple countries of origin.

**Table 2.** Participant group averages with standard deviations.

| Variable | SDBA | | BES-7/8 | | ME-7/8 | | BES-5 | | MLE-5 | |
|---|---|---|---|---|---|---|---|---|---|---|
| | μ | SD | μ | SD | μ | SD | μ | SD | μ | SD |
| Frequency of use of Spanish (max. 30 points) | 15.2 | 5.9 | 15.7 | 4.7 | 14.0 | 4.6 | 15.5 | 6.2 | 13.7 | 4.2 |
| Proficiency score (max. 18 points) | 16.1 | 1.9 | 14.4 | 2.9 | 13.9 | 3.3 | 11.2 | 4.3 | 14.4 | 2.9 |
| Number of monolingual Spanish-speaking parents | 1.9 | 0.2 | 1.0 | 0.9 | 1.4 | 0.8 | 0.9 | 0.9 | 1.4 | 0.9 |

*3.2. Methods*

The experiment was administered using Qualtrics, whereby SDBA participants completed activities asynchronously and HSs carried out all tasks in their schools using laptop computers in the presence of the researcher. All participants completed an 18-question segment of the Bilingual English–Spanish Assessment (BESA; Peña et al. 2018) to measure morphosyntactic proficiency in Spanish, which tested determiner-noun gender and number agreement ($k = 4$), verbal person/number agreement ($k = 4$), preterit aspect ($k = 2$), clitic gender and number agreement ($k = 4$), and subjunctive mood ($k = 4$). In addition, they completed a brief language questionnaire that identified which members of their family spoke Spanish as well as the frequency with which they used Spanish in six contexts (with parents, with other family members, with friends, at school, in public, while watching television) rated on 1–5 Likert scales. These data allowed for addressing individual differences in frequency of use and proficiency as continuous variables.

In addition to the BESA and language questionnaire, participants carried out two experimental tasks. The first was the sentence completion task (SCT), which targeted the oral production of DOM; the second was the morphology selection task (MST), which tapped participants' receptive knowledge of this structure. The two tasks used the same eight transitive verbs to avoid the possibility that differences between the sets of lexical items used on each could have influenced results. All verbs were morphologically regular and disyllabic and ended in –ar, which is the most common of the three conjugation classes in Spanish. The subject of all clauses was plural (*las hermanas*, 'the sisters'), which increased the salience of DOM.[6] Both tasks were situated within a communicative context in which a mother describes her desires for her children while they are away at sleepaway camp. Appendix A contains example stimuli from the SCT and MST.

The children's SCT contained ten target items and six distractors. The adults' SCT contained the same items, as well as 31 additional distractors.[7] There was also a practice item. Each stimulus contained a brief description followed by an incomplete sentence that began with a matrix clause with either the verb *querer* ('to want', which takes a subjunctive complement) or *creer* ('to believe', which takes an indicative complement) as part of a larger experiment exploring HSs' mood systems. Participants needed to record their voice completing the sentence using a form of the verb whose infinitive was provided in parentheses, as well as any other words. These instructions and the practice item were conducive to the suppliance of the dative marker *a*, which was intentionally omitted in the subordinate clause.

On the MST, there were eight stimuli targeting the use of DOM with the phrase *tienen que* ('they have to') and the infinitive in question. In addition to the eight target items, the

children completed fifteen distractor items; the adults completed 24 additional distractors. The MST followed a similar structure to the SCT and featured the same communicative focus; however, participants needed to read prompts and then select which of two sentences looked best to them. One sentence contained the differential object marker *a* between the verb and the direct object, *Juanito*, and one omitted it. The verb and the direct object *Juanito* (as well as the differential object marker, where present) were placed in bold to highlight the contrast between the two sentences. The answer containing DOM was alternated between the first and second choice to prevent the responses from becoming predictable.

## 4. Results

All data analyses took place in RStudio (R Core Team 2022) using the *emmeans* (Lenth 2021), *lme4* (Bates et al. 2015), *lmerTest* (Kuznetsova et al. 2017), *sjPlot* (Lüdecke 2021), and *tidyverse* (Wickham et al. 2019) packages. For all descriptive, inferential, and individual analyses, the production or selection of DOM served as the binary dependent variable. In all instances where participants produced or selected DOM, the response received a score of 1, and omission resulted in a score of 0. The production of alternative but grammatical forms (e.g., *quiere que cuiden de Juanito*, 'she wants for them to care for Juanito', in which both the dative marker *a* and the preposition *de* are both grammatical) were removed from the analysis, as were sentences where participants' recordings were not saved or were incomprehensible. This left 672/750 (89.6%) observations in the production data for analysis. Anonymized data are available on the GitHub repository listed in the data availability statement below. Each participant group's rates of DOM production and selection are summarized in Figures 1 and 2.

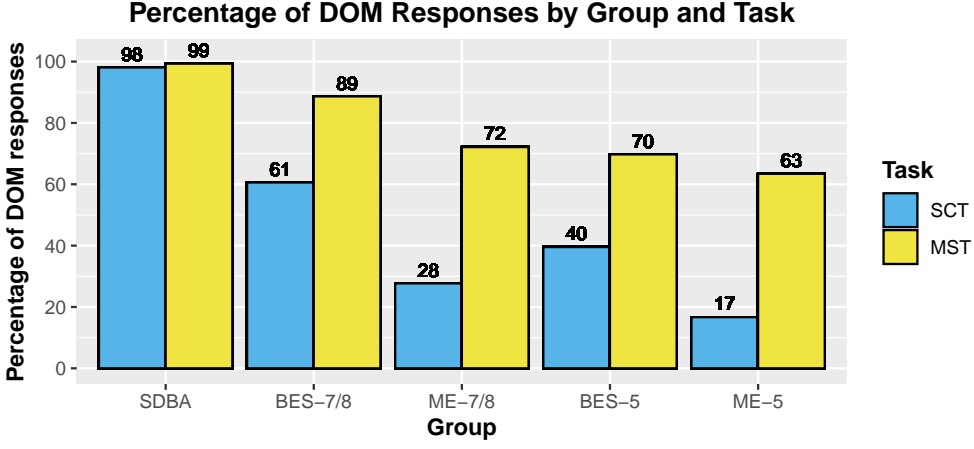

**Figure 1.** Average rates of DOM production and selection by group.

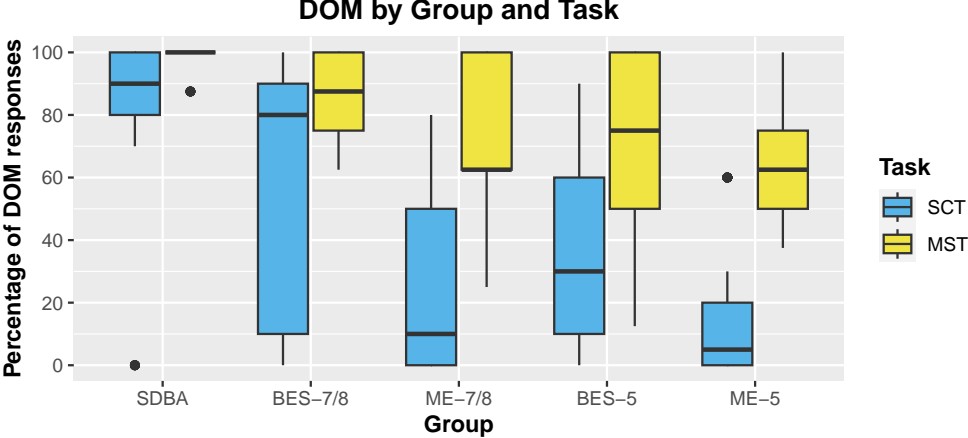

**Figure 2.** Statistical summary of percentages of DOM production and selection by group.

The independent variables for the present analysis were the five-way categorization of participant group (SDBA, BES-7/8, ME-7/8, BES-5, ME-5), school (DLI versus monolingual English), age group (fifth grade versus seventh/eighth grade), task (SCT versus MST), frequency of use, and BESA proficiency score. The participant group, school, and age group variables were categorical, and frequency of use (sum of Likert scales across six contexts of use; maximum 30) and BESA proficiency score (number of correct responses; maximum 18) were continuous variables that were standardized prior to analysis. The effect of frequency of use on DOM production and selection is summarized in Figure 3, and that of proficiency is summarized in Figure 4.

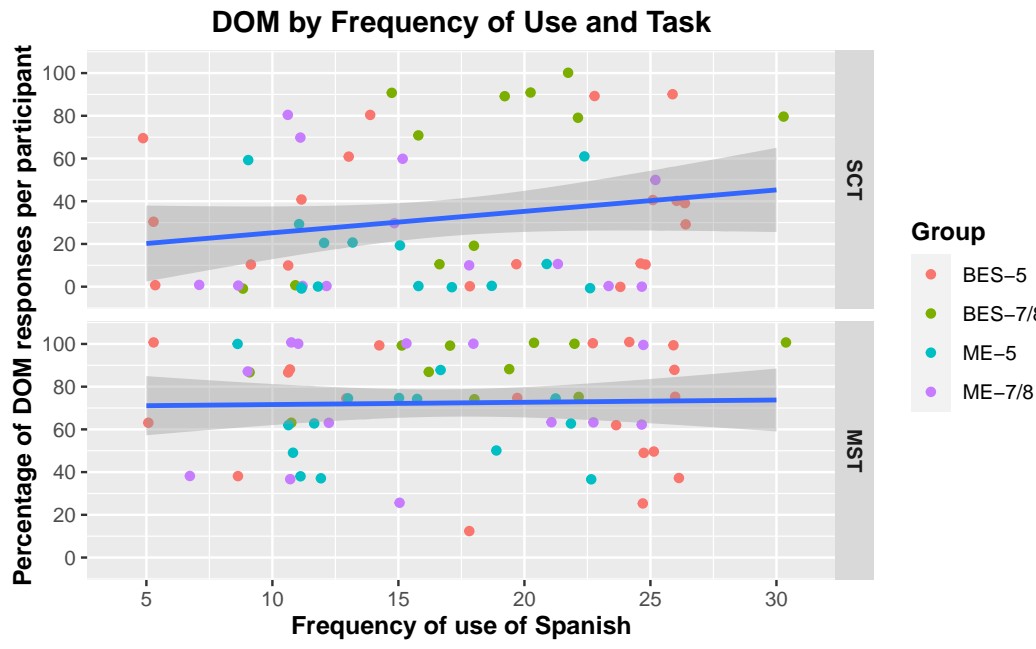

**Figure 3.** Individual rates of DOM production and selection by frequency of use.

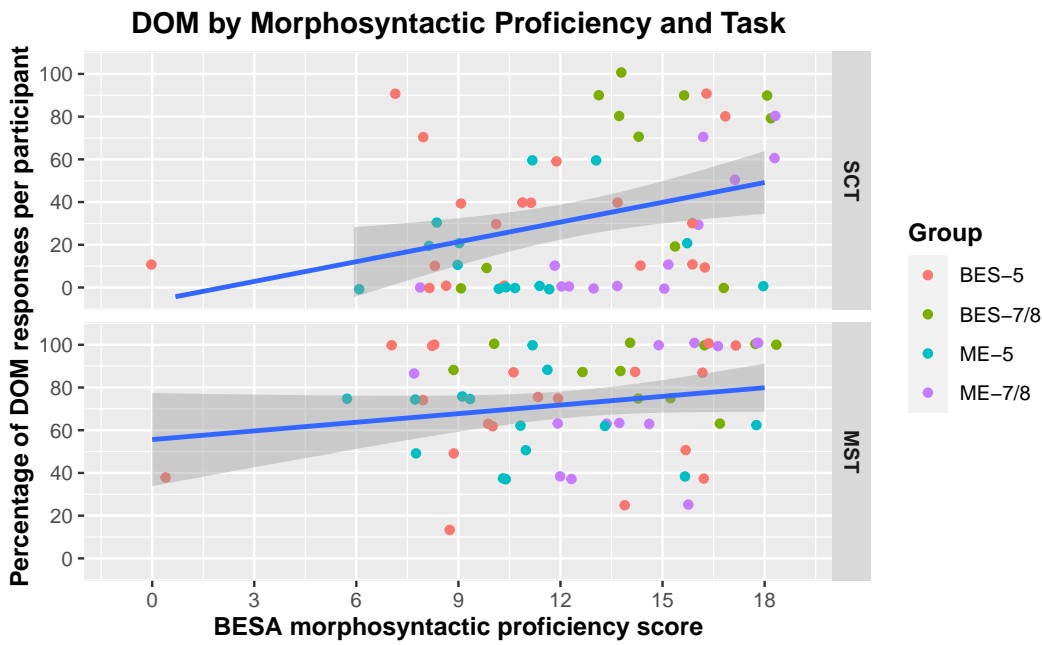

**Figure 4.** Individual rates of DOM production and selection by BESA proficiency.

*4.1. Inferential Statistics*

To further explore these results, four Generalized Linear Mixed Methods (GLMM) binomial logistic regression models were necessary. DOM production and selection was the binary dependent variable, and participant and item were random effects for all models. In the first model, all participants' data were included to compare the HSs to SDBAs. In the final three models, only the HSs' data were incorporated to explore variability within this heterogeneous group. In the first model, which included data from both tasks and all groups, participant group and task were the dependent variables. The SDBA group was set as the reference level for group and the SCT as the reference level for task. In the resulting model, there were main effects significant at the $p < 0.05$ level for all of the predictors, as summarized in visual form in Figure 5: BES-7/8 group ($\beta = -4.01$, SE = 0.87, $p < 0.001$), ME-7/8 group ($\beta = -5.98$, SE = 0.85, $p < 0.001$), BES-5 group ($\beta = -5.60$, SE = 0.81, $p < 0.001$), ME-5 group ($-6.63$, SE = 0.85, $p < 0.001$), and MST ($\beta = 2.46$, SE = 0.38, $p < 0.001$). All group effects were negative, implying quantitative differences in DOM production and selection between HSs and the SDBAs; the MST favored the use of DOM over the SCT.

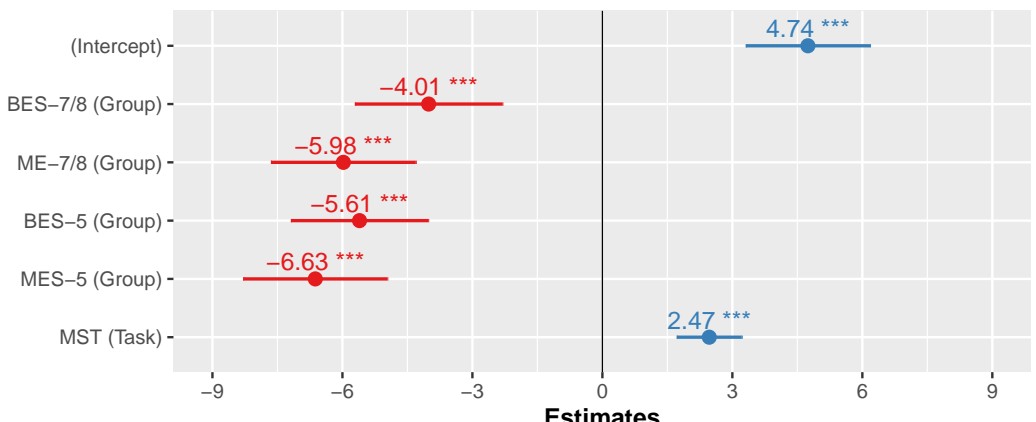

**Figure 5.** Results of GLMM binomial logistic regression for participant group and task. Asterisks indicate level of statistical significance, with *** indicating the greatest significance.

Since one of the primary goals of this study was to explore which factors account for how HSs differ from one another, Tukey post-hoc comparisons between the HS groups were necessary. The differences between the BES-7/8 and ME-7/8 groups ($\beta = 1.97$, SE = 0.72, $p = 0.047$) and the BES-7/8 and ME-5 groups ($\beta = 2.62$, SE = 0.71, $p = 0.002$) were significant at the $p < 0.05$ level, implying that the older children from the DLI school showed stronger knowledge of DOM than both groups of participants who attended the monolingual school. There were no differences significant at the $p < 0.05$ level between children in the monolingual school at either grade level or between the fifth grade students in either school.

To better address individual variability in the HSs' data across tasks, a second GLMM model was prepared with school, age group, task, frequency of use, BESA proficiency score, and the school-by-age group interaction as fixed effects. The DLI school served as the reference level for school, fifth grade as the reference level for age group, and SCT as the reference level for task. The model revealed significant effects at the $p < 0.05$ level for the seventh/eighth grade age group ($\beta = 1.58$, SE = 0.63, $p = 0.011$), the MST ($\beta = 2.47$, SE = 0.36, $p < 0.001$), and BESA proficiency ($\beta = 0.49$, SE = 0.22, $p = 0.025$), as summarized in visual form in Figure 6.

**Summary of GLMM Model for Heritage Speakers**

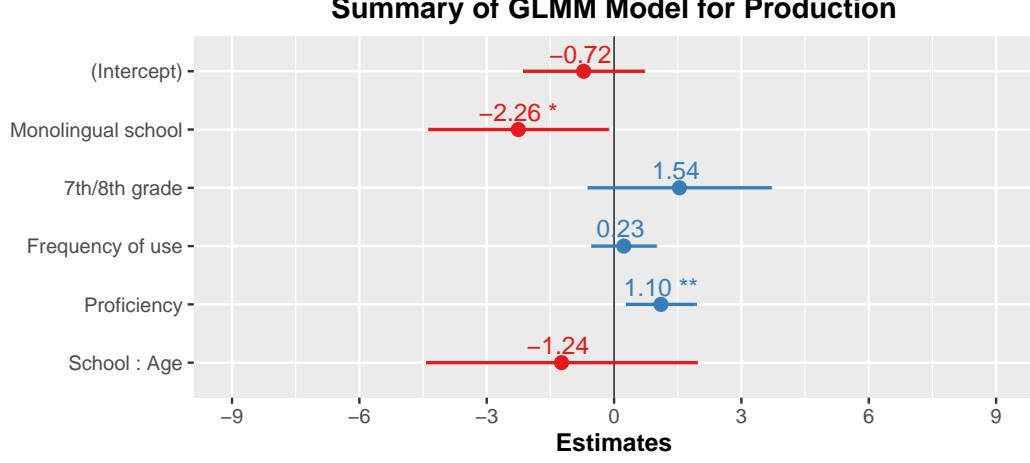

**Figure 6.** Results of GLMM binomial logistic regression for HSs across tasks. Asterisks indicate level of statistical significance, with *** indicating the greatest significance.

Finally, two additional GLMM models were necessary to evaluate the same effects within each task. The third GLMM model incorporated HSs' data from the SCT only, and the fourth GLMM model incorporated HSs' data from the MST only. With the exception of task, the same predictors as the previous model—school, age group, frequency of use, BESA proficiency, and the school by age group interaction—were included as fixed effects. Once again, the DLI school served as the reference level for school, and fifth grade as the reference level for age group. In the SCT model, which is summarized in Figure 7, there were main effects significant at the $p < 0.05$ level for the monolingual English school ($\beta = -2.26$, SE = 1.08, $p = 0.037$) and for BESA proficiency ($\beta = 1.10$, SE = 0.42, $p = 0.008$). In the MST model, which is summarized in Figure 8, there was a main effect significant at the $p < 0.05$ level for the seventh/eighth age group only ($\beta = 1.44$, SE = 0.63, $p = 0.022$).

**Summary of GLMM Model for Production**

**Figure 7.** Results of GLMM binomial logistic regression for HSs on the SCT. Asterisks indicate level of statistical significance, with *** indicating the greatest significance.

To summarize the inferential statistics, the SDBAs produced and selected DOM at ceiling in the expected contexts, which was more than all groups of HSs, who were more likely to select this structure on the MST than to produce it. The BES-7/8 group produced and selected the dative marker *a* more frequently than all students in the monolingual school, and there were no differences between children in the fifth grade groups or between children in the two monolingual school groups. Overall, age group and proficiency level modulated HSs' performance. However, DOM production was modulated by school and

proficiency, while age group accounted for variability in selection. These data suggest that an interplay between age and type of schooling affects knowledge of DOM in Spanish HSs, depending upon whether production or receptive knowledge is involved. Current exposure as operationalized through proficiency represented individual differences between HSs, especially in production, while cumulative exposure as represented through age affected underlying knowledge.

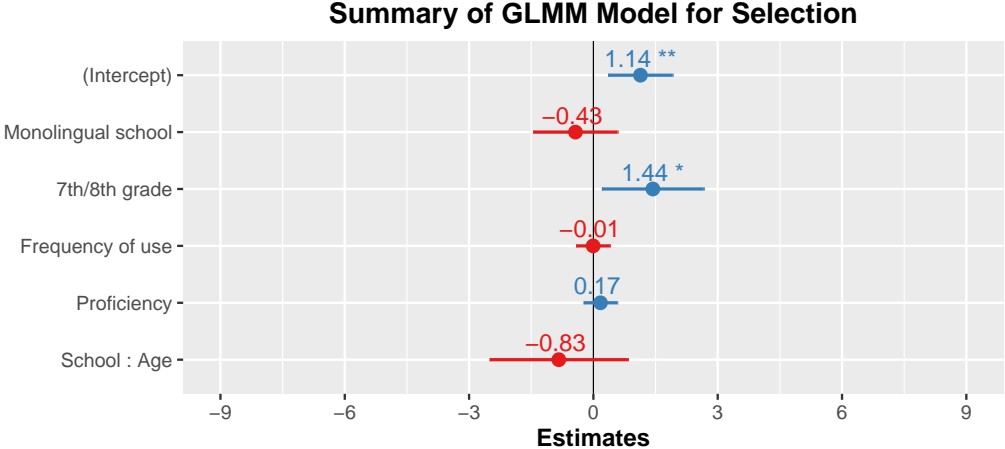

**Figure 8.** Results of GLMM binomial logistic regression for HSs on the MST. Asterisks indicate level of statistical significance, with *** indicating the greatest significance.

### 4.2. Individual Analyses

Turning now to individual analyses, each HS's amount of production and selection of DOM is represented in Figure 9. Note that all HSs used DOM in at least 1/18 possible contexts, and all but one participant did so in at least 3/18 contexts, arguing against the altogether absence of this structure from any of their grammars. Virtually all participants selected this structure more than they produced it, which is in line with the predictions and the statistical modeling.

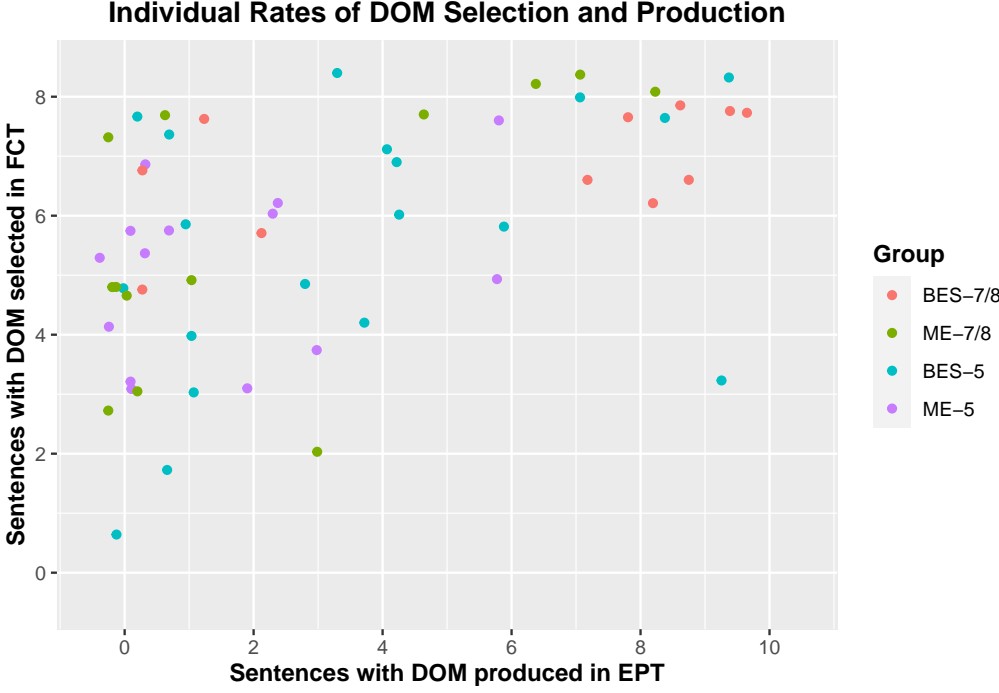

**Figure 9.** Results of GLMM binomial logistic regression for HSs across tasks.

All SDBAs used DOM in at least 16/18 instances. One HS produced and selected DOM in all 18/18 sentences, and eight HSs converged on the range of variability observed in SDBA participants' DOM system. The characteristics of the eight HS participants who converged on the production rates of the SDBAs are summarized in Table 3. 7/8 attended the DLI school, and 6/8 were in the older participant group (1/6 in the seventh grade, 5/6 in the eighth grade). Except for participants H5B08 and H8B09, these participants approached ceiling in their performance on the BESA proficiency test of morphosyntax. These findings align with the inferential statistics and reveal that almost all children whose DOM rates converged on those of the group that represents their input attended the DLI school. No patterns in frequency of use or parental language background emerged across these participants.

**Table 3.** Summary of background factors for HS participants who produced and selected DOM within the range of the SDBA group.

| Part. | DOM | Group | Freq. Use[8] | School Use | BESA | Parent Langs. |
|-------|-----|-------|-----------|------------|------|---------------|
| H8B04 | 18/18 | BES-7/8 | 19/25 | 3/5 | 14/18 | 2 monolingual Spanish |
| H5B08 | 17/18 | BES-5 | 19/25 | 4/5 | 7/14 | 2 bilinguals |
| H7B02 | 17/18 | BES-7/8 | 17/25 | 3/5 | 18/18 | 1 bilingual, 1 monolingual Spanish |
| H8B06 | 17/18 | BES-7/8 | 13/25 | 2/5 | 16/18 | 2 bilinguals |
| H5B19 | 16/18 | BES-5 | 12/25 | 2/5 | 17/18 | 1 bilingual, 1 monolingual Spanish |
| H8B08 | 16/18 | BES-7/8 | 25/25 | 5/5 | 18/18 | 2 monolingual Spanish |
| H8B09 | 16/18 | BES-7/8 | 16/25 | 3/5 | 13/18 | 2 bilinguals |
| H8M02 | 16/18 | ME-7/8 | 11/25 | 0/5 | 18/18 | 1 bilingual, 1 monolingual Spanish |

In sum, these individual data highlight that HSs have gradient knowledge of DOM: only one participant produced this structure at ceiling, and there was not a single participant who did not produce or select DOM in any of the contexts. Those speakers who converged on the norms of the SDBAs were primarily enrolled in the DLI school, had high proficiency in Spanish, and were in the older participant group. Participants selected sentences with DOM more than they produced this form. These findings align with the descriptive and inferential statistics but show that at the individual level, there is a wide range of variability in DOM use.

## 5. Discussion

The present study contributes to a growing body of research that focuses on the dynamic differences between HSs and addresses HL acquisition and change within a feature-oriented framework (Putnam and Sánchez 2013). Through testing DOM in both the productive and receptive domains in bilingual children and adolescents aged ten to fourteen years, this study makes inroads into elucidating the individual factors that shape HL acquisition during this understudied developmental period. Furthermore, this study adopts a novel approach to the acquisition of Spanish as a HL by evaluating how DLI education contributes to acquisition compared to traditional schools. Participants completed production and selection tasks that tested their knowledge of DOM, a structure that has proven to be variable in child and adult HSs of Spanish. Even monolingual children show (lower degrees of) variability in the acquisition of DOM (Guijarro-Fuentes and Marinis 2011; Guijarro-Fuentes et al. 2017), so exposure to Spanish through DLI may be particularly important for sustaining the development of this structure. In so doing, this study contributes to the recent "paradigm shift" (Giancaspro et al. 2022, p. 484) away from monolingual normativity and towards a more nuanced understanding of the experiential factors that shape individual variability in the acquisition of Spanish as a HL

and distinguishes this group from other bilinguals (see Rothman et al. 2023 in support of comparing HSs to other HSs and Pascual y Cabo and Rothman 2012 for an argument in favor of bilingual comparison groups).

The first research question evaluated whether HSs enrolled in a DLI school, which offered its students 50% of academic instruction in Spanish during the primary years, would produce and select similar levels of the dative marker *a* with animate and specific direct objects when compared to SBDAs and age-matched HSs in a monolingual English school. It was predicted that SDBAs and HSs in the DLI school would produce and select the dative marker *a* with animate and specific direct objects at ceiling, more than HSs who receive a traditional English-only education. The SDBAs produced DOM at ceiling, while the HSs groups did not, which is in partial alignment with the hypotheses. The BES-7/8 participants, the older children who had attended a DLI school, were more likely to produce and select DOM than other children. This implies that age and education in Spanish interact in the HL acquisition of this structure; however, the effect of school exposure is only significant at the $p < 0.05$ level in the production, but not in recognition, of DOM, and no interaction between age and school was found in either task. Notably, in production, the HSs who attended the DLI program produced twice as much DOM than participants in the monolingual English-only school.

This argues that the effect of bilingual education has a greater impact on the production of DOM than receptive knowledge. Intuitively, children in DLI produce texts and interact in Spanish more frequently than monolingually educated children. However, previous studies have shown that HSs use English frequently during Spanish instruction (Babino and Stewart 2017; Hamman 2018; Potowski 2004). Moreover, frequency of production in the HL is deterministic in young children's command of Spanish (see Goldin 2021 regarding subject–verb agreement and Sánchez et al. 2023 concerning null subjects), such that those students who produce in Spanish more frequently show stronger knowledge of morphosyntactic structures. Together, these findings suggest that DLI does provide participants with the chance to produce in the HL, which has a material impact on acquisition in the present study. However, it is likely that those participants who produce more frequently in Spanish benefit more from DLI than others. If so, this implies that input is a necessary but insufficient condition for HL maintenance, as argued in second language acquisition research (e.g., Swain 1993), which is a claim that is underscored by the productive-receptive asymmetry in this study. Future research that measures the frequency of input and output separately is necessary to disentangle the relative impact of each.

The second research question targeted the roles of frequency of use and proficiency, which comprise proxies to current HL exposure (López Otero and Jimenez 2022; Giancaspro and Sánchez 2021), on individual HSs' rates of production and selection of DOM with animate and specific direct objects. In alignment with previous research on DOM with bilingual children and adults, it was anticipated that both would account for the production and selection of this structure. However, only proficiency affected speakers' knowledge, particularly in production. This finding is still compatible with Putnam and Sánchez's (2013) approach to HL acquisition that emphasizes experience with and exposure to the HL, as they argue that production is more susceptible to exposure effects than receptive knowledge. This finding also supports Sánchez's (2019) claim that HSs may have a bilingual alignment that reflects innovative production tendencies due to patterns of use and exposure while underlying receptive knowledge remains (more) stable. However, the absence of an effect for frequency of use in any of the statistical models differs from previous experiments that have found that this variable accounts for differences in language use between speakers (e.g., Dracos and Requena 2022; Martínez Vera et al. 2023; Montrul and Sánchez-Walker 2013; Perez-Cortes 2016; Solano-Escobar and Cuza 2023).

The third research question assessed whether age would modulate participants' command of Spanish DOM with animate and specific direct objects. It was hypothesized that participants in the seventh/eighth grade group would produce and select DOM in this context more consistently than HSs in the fifth grade, which was observed in the

overall findings and the receptive data, but not in production. These findings differ from Guijarro-Fuentes and Marinis' (2011) and Guijarro-Fuentes et al.'s (2017) studies with teenagers, since the growth in the present study took place at a similar age range as that of the participants in their experiment.

These findings align with some previous studies on other areas of the Spanish inflectional system that have found a facilitative effect for age (Corbet and Domínguez 2020; Cuza and Miller 2015; Martinez-Nieto and Restrepo 2022; Montrul and Potowski 2007; see also Flores et al. 2017 concerning European Portuguese HSs), which points to protracted HL development of inflectional morphology. These studies differ, however, from some previous work on child Spanish HSs that reports attrition of morphosyntactic structures (Goebel-Mahrle and Shin 2020; Merino 1983), as well as the predictions that Putnam and Sánchez (2013) advance, since it does not appear that bilingual children are restructuring their grammatical repertoire over time. This finding is particularly supported by the fact that, in particular, the seventh/eighth grade children in the DLI school showed stronger knowledge of DOM than other participants, yet these bilinguals had received a drop in exposure to Spanish after exiting their immersion program in the fifth grade. Consequently, early immersive exposure may be a facilitative ingredient for acquiring the differential object marker in heritage varieties of Spanish, but even still, this process appears to continue into early adolescence and may continue to show quantitative differences from SDBAs.

Future studies would benefit from incorporating younger and older groups of children to better plot the course of development of this structure into adulthood. In the absence of such data, it is prudent to look to other studies concerning input effects with bilingual children. One speculative possibility is that the Spanish HS children who attended the monolingual school would, at later ages, converge on the production rates of the BES-7/8 group. This would imply that there exists a relationship between age and exposure, such that those bilinguals who receive more HL input are able to acquire structures faster than others, which is a finding documented previously in Portuguese HSs' acquisition of the subjunctive mood (Flores et al. 2017). If this is the case, DLI affects the *rate* of acquisition. Alternatively, it is possible that the children who attended the monolingual school will not show further growth at the group level in their acquisition of DOM, pointing towards an advantage for children who receive bilingual education in the ultimate *route* of HL acquisition. Neither possibility precludes subsequent attrition, but both point to an advantage for DLI in the acquisition of Spanish as a HL during the school years.

The fourth and final research question asked whether the child HSs in this study would show asymmetrical productive and receptive knowledge of DOM with animate and specific direct objects. In alignment with the asymmetries observed when comparing Guijarro-Fuentes and Marinis' (2011) production data with the receptive tasks reported in Guijarro-Fuentes et al. (2017), and in conformity with Putnam and Sánchez's (2013) predictions, it was hypothesized that HSs would select DOM more than they would produce this structure. This finding is supported by descriptive, inferential, and individual analyses. The effect favoring the MST shows that HSs are more readily able to integrate their syntactic and semantic knowledge of DOM for receptive purposes than to activate this structure in production. This is particularly true for those HSs with lower proficiency in and activation of Spanish, since proficiency, a proxy for current exposure, accounted for individual differences in production only. The fact that children in the DLI school, who likely have greater opportunities to produce in (and therefore activate) Spanish, were more likely to produce DOM also supports this conclusion. These claims support the frameworks of bilingual alignments (Sánchez 2019) and differential access (Perez-Cortes et al. 2019) since current exposure (as defined by proficiency level) affects access to HL grammatical knowledge in production more than at the receptive level, where the acquisition of DOM is modulated by cumulative exposure (as defined by age).

The present study largely aligns with previous research on other HLs that have exposed that literacy and education facilitate HL acquisition (Bayram et al. 2017; Kupisch et al. 2014; Kupisch and Rothman 2018). Nevertheless, despite having received up to six

years of instruction with 50% of Spanish throughout the school day, even the BES-7/8 participants, who showed the strongest knowledge of DOM, did not converge on the system of adult bilinguals who represent their input in Spanish. This differs from previous findings in which many or all HSs with high literacy or who had been educated in the HL converged on the syntactic systems of speakers dominant in the same language: 8/58 HSs (13.7%) in this study were within the range of the SDBAs in the use of DOM across tasks. Consequently, while the present data show that DLI provides an *advantage* for the HL acquisition of DOM, they do not completely mitigate crosslinguistic influence to the same extent that is observed in other studies reporting effects for HL literacy and bilingual education (e.g., Bayram et al. 2017; Kupisch and Rothman 2018). Kupisch and Rothman (2018) argue that broader sociolinguistic contexts condition the impact of bilingual schooling and that the United States does not place emphasis on bilingualism more generally. This in turn emphasizes the importance of contextual factors surrounding HL acquisition, and that there is no one-size-fits-all approach that can account for all HSs without considering social and experiential factors.

The data from the present study also largely (but not entirely) support Putnam and Sánchez's (2013) feature-oriented approach to HL acquisition and maintenance. This model correctly predicts a role for patterns of exposure, particularly in production, a finding that is supported (A) through the effect of exposure to Spanish via bilingual education and (B) by the effect of proficiency, a proxy for current exposure (Giancaspro and Sánchez 2021; López Otero and Jimenez 2022). Additional evidence for this framework stems from the finding that HSs recognized and selected DOM more often than they produced it.

However, the positive correlation between age and DOM production and selection does not support a strict interpretation of Putnam and Sánchez's (2013) predictions. Specifically, these researchers posit that HSs will restructure their grammatical knowledge over time rather than show quantitatively greater use of a particular structure. A tentative claim that needs further evidence is that this approach to HL feature reassembly runs in reverse to account for the protracted development of inflectional morphology in bilingual children. Specifically, this would imply that children continue to acquire functional features such as DOM as their cumulative amount of input in the HL increases, particularly at the receptive level, and then begin to produce DOM more consistently after prolonged and sustained HL exposure. The data presented here support this claim, whereby the majority of growth across groups transpires in production (see Figure 1). This emphasizes the importance of both cumulative and current exposure to the HL in the acquisition of morphological and syntactic structures such as DOM. The claim that Putnam and Sánchez's (2013) predictions are bidirectional has already been advanced in research with adult bilinguals (Thane 2023). Regardless of the theoretical framework subsumed, it is important to note that all of the HSs in this study produce and/or select DOM variably (or categorically), such that the underlying syntactic knowledge of this structure seems to be intact and is not altogether unacquired.

Although Putnam and Sánchez's (2013) approach to HL acquisition has been essential in exposing the myriad factors that can shape the outcomes of bilingual development, the data here are also compatible with accounts of language acquisition that posit quantitative differences between HSs' and SDBAs' grammatical knowledge (e.g., Montrul 2008, 2013). Putnam and Sánchez's (2013) framework has facilitated the development of multiple predictions related to language experience that have been tested here to provide a more fine-grained account of variability between different HSs. Such an approach has allowed this and other studies in the present volume that have concentrated on experiential factors (e.g., López Otero 2023; Martínez Vera et al. 2023; Solano-Escobar and Cuza 2023) to elucidate the interplay of variables that give shape to HL acquisition, which is a "paradigm shift" (Giancaspro et al. 2022, p. 484) in approaching HL research. As shown in an analysis of the individual data, some HSs converge on the ranges of SDBAs, and multiple factors such as bilingual education, patterns of current exposure, and age conspire to account for within-group differences that are not addressed within traditional approaches to HL

acquisition. The present study also contributes to this body of work the finding that DLI impacts the acquisition of areas of the Spanish morphosyntactic system that are often variable in other HS populations, although it is undeniable that at the group level, HSs do differ from other populations of Spanish speakers in their command of DOM at both the productive and receptive levels even after receiving bilingual education.

Putnam and Sánchez (2013) also make predictions that could account for HSs' gradient knowledge of DOM that were not tested in the present study. One possibility that would align with these authors' proposal is that HSs are sensitive to the lexical frequency of the verbs preceding the differential object marker. Another possibility follows the argument advanced by von Heusinger and Kaiser (2007) that some verbs are more likely to take animate and specific direct objects than others. Working within Jiang's (2000) lexical approach to language acquisition, Hur (2021) found that both such factors affected adult HSs' use of DOM, so both are variables that may hold promise in accounting for gradient variability in future studies with bilingual children. DLI may minimize frequency effects compared to other contexts of HL acquisition for two reasons. Firstly, this context purportedly provides HSs with exposure to a greater variety of lexical items that are typically specific to academic registers. Secondly, DLI programs may integrate form-focused instruction on specific structures that could promote their use in ways that are not restricted to individual lexical items (form-focused instruction in DLI programs is also encouraged by Harley 1993; Snow et al. 1989). Future work may wish to test this claim.

Notwithstanding these contributions to the field, this study has multiple limitations that form the foundation for future research. For instance, as mentioned previously, this study may have benefitted from additional participant groups who were younger and older than those tested here to better understand the progression of the development of DOM in Spanish as a HL. Given the lengthy nature of the experiment, testing younger age groups may have been problematic, but would hold promise in further substantiating the claims that HS children do not attrite, but rather continue to acquire, the DOM system over the school years and into adulthood. Moreover, the present study did not test instances of DOM commission, whereby child HSs may overextend the differential object marker into inanimate and/or nonspecific contexts, which has been found with bilingual children (Callen 2023; Cuza et al. 2019) and in monolingual communities (Sánchez and Zdrojewski 2013; von Heusinger and Kaiser 2005, 2007). Lastly, the present study evaluated the production of mood and DOM in subordinate clauses using the same stimuli but tested these structures through separate, and therefore less complex, stimuli on the MST. While the findings of task asymmetries align with the predictions and with linguistic theory, future research that targets DOM production in specific stimuli would be useful.

Before closing, although the present study has made theoretical contributions to the study of Spanish as a HL, it is important to think about the broader implications of these data for DLI programs. For instance, the data from the present study found a role for bilingual education in the production of DOM, but there is still considerable variability within the DLI groups in their production and receptive knowledge of this structure. Therefore, a greater concentration of Spanish instruction in DLI programs would be beneficial, and sustaining immersion across the secondary years may further support the acquisition of syntactic structures such as DOM. DLI programs should also consider integrating form-focused grammar instruction into Spanish language arts courses. Previous research has shown that explicit instruction may be necessary for HSs to acquire other areas of inflectional morphology such as the subjunctive mood (Montrul and Perpiñán 2011) to the same level as schooled learners such as advanced second language speakers. Moreover, the fact that children showed growth in their command of DOM after having exited an immersion program suggests that this method of education provides some degree of protection against HL attrition, although it is unclear if these HSs would continue to conserve their knowledge of DOM at later stages.

Future studies on DLI are necessary, as they allow those of us who are committed to promoting initiatives to maintain multilingualism to achieve social justice for multilingual

speakers. The impact of DLI goes well beyond HL acquisition itself. In addition to honoring the linguistic rights of HSs (Skutnabb-Kangas 2006), DLI has been shown to benefit Hispanic students' academic performance as measured using standardized assessment scores (e.g., Lindholm-Leary and Borsato 2005; Marian et al. 2013; Serafini et al. 2020) and is at least as effective in fostering the acquisition of English (e.g., Acosta et al. 2019; Marian et al. 2013; Umansky and Reardon 2014). DLI also expedites the development of executive functioning skills (e.g., Garraffa et al. 2020; Nicolay and Poncelet 2015). Hispanic children in DLI programs participate in an educational experience that validates their identities as speakers of both languages of instruction and have positive attitudes towards bilingualism and biculturalism (de Jong and Bearse 2011). Nevertheless, future work may wish to concentrate on the best pedagogical practices that reinforce linguistic structures that are used in lessons across the content areas (Harley 1993; Snow et al. 1989). DLI programs offer an opportunity for teaching language through content *and* form-focused approaches to maximize HL development. Therefore, providing an even stronger context for the acquisition of Spanish is an ongoing goal for these programs.

## 6. Conclusions

The present study contributes to a growing body of recent research that concentrates on experiential variables in the acquisition of Spanish as a HL. Specifically, this project shows that bilingual education and exposure (both current, as reflected by proficiency, and cumulative, as seen through the effect of age) facilitate English-dominant Spanish HS children's acquisition of DOM. HS children were more likely to select DOM than they were to produce this structure. Having attended a bilingual school accounted for variability in production in particular, along with proficiency, yet cumulative exposure as represented by age accounted for differences between participants at the underlying receptive level. These findings constitute additional evidence for Putnam and Sánchez's (2013) framework, although the directionality of predictions that HSs will restructure their knowledge does not line up with the process of protracted development that is portrayed by the data from the present study. The findings in the individual analyses indicate that all children who converged on the range of variability of the SDBA participants were enrolled in the bilingual school and/or in the older age group, further supporting the importance of these factors in the acquisition of the morphological and syntactic system of Spanish as a HL.

These findings suggest that taking language experience into consideration, including bilingual education, current exposure, and age, is essential in HL acquisition research. This study is the first to expose the effects of bilingual education on HS children's production and receptive knowledge of Spanish and builds upon the findings within this special issue that argue that sustained HL exposure is essential to foster the development and maintenance of structures that may otherwise become restructured or differentially acquired by HSs. This in turn sets the stage for future research that works to compare HSs to one another rather than exclusively to other types of bilinguals.

**Funding:** This research received no external funding.

**Institutional Review Board Statement:** The study was conducted according to the guidelines of the Declaration of Helsinki, and approved by the Institutional Review Board of Rutgers University (protocol code Pro2021001902, most recent amendment approved 11 March 2022).

**Informed Consent Statement:** Informed consent was obtained from all subjects involved in the study.

**Data Availability Statement:** Please see the following GitHub repository for code and analysis: https://github.com/pthane/DLI-Morphosyntax-2023.

**Acknowledgments:** This project is dedicated to Phoebe Ann Search (1978–2022), who overcame adversity many times yet never stopped spreading joy for life, passion for education, and commitment to social justice. She epitomizes the role of an inspiring educator. Not all heroes wear capes. A thank you also goes to her faithful teaching partner and friend, Dot Cates, for carrying on her memory. In

addition to the anonymous reviewers, I would like to thank Alejandro Cuza for his enthusiasm for publishing my manuscript in this special edition, as well as Jennifer Austin, Julio Cesar López Otero, and David Giancaspro for their continued and valuable feedback at various stages of this project.

**Conflicts of Interest:** The author declares no conflict of interest.

## Appendix A  Sample Stimuli

Sample stimulus from SCT with the matrix verb *querer* ('to want'):

> A veces Juanito se pone triste si sus hermanas dicen que no quieren hablar con él. ¿Qué quiere la mamá? Quiere que las hermanas _________ (LLAMAR) Juanito cada noche.
>
> *Sometimes Juanito gets sad when his sisters say that they don't want to talk with him. What does the mom want? She wants the sisters _________ (TALK; subjunctive inflections + DOM expected) Juanito every night.*

Sample stimulus from SCT with the matrix verb *creer* ('to believe"):

> Los tres hermanos tienen muy buena relación. ¿Qué cree la mamá? Cree que las hermanas _________ (AMAR) Juanito.
>
> *The three siblings have a great relationship. What does the mom believe? She believes that the sisters _________ (LOVE; indicative inflections + DOM expected) Juanito.*

Sample stimulus from MST with the matrix phrase *tienen que* + infinitive ('have to' + infinitive):

> ¿Qué tienen que hacer las hermanas?
>> a. Tienen que **llamar a Juanito** cada día.
>> b. Tienen que **llamar Juanito** cada día.
>
> *What do the sisters need to do?*
>> *a. They need to call (with DOM) Juanito every day.*
>> *b. The need to call (no DOM) Juanito every day.*

## Notes

[1]  Putnam et al. (2019, p. 19) claim that features are "indices on lexical items and larger syntactic objects that allow generated structures to be interpreted at external interfaces".

[2]  The use of the dative marker *a* is also contingent upon additional semantic and pragmatic constraints, such as topicality, lexical aspect of the preceding verb, subject agentivity, and definiteness of the object (Fábregas 2013; Torrego 1998; Zagona 2002). DOM also occurs in some sentences where both the subject and object are inanimate to differentiate between them (Rodríguez-Mondoñedo 2008). The present study concentrated only on animate and specific direct objects, the "core" case of DOM (Aissen 2003), so discussion is limited to this context only.

[3]  As stated previously, multiple recent studies have argued that proficiency represents participants' levels of HL exposure at the time of testing (Giancaspro and Sánchez 2021; López Otero and Jimenez 2022).

[4]  Although Reina et al. (2021) document the retraction of DOM in Caribbean varieties of Spanish, there is no evidence that Dominican speakers omit the dative marker *a* with definite nouns that are animate and specific. Furthermore, Aissen (2003) argues that proper nouns such as the ones elicited in this experiment are maximally animate and specific and are most likely to result in the use of DOM crosslinguistically. Finally, the data from the Spanish-dominant bilingual adult comparison group, which included Caribbean speakers, revealed ceiling-level production of DOM in the expected contexts, arguing against the role of dialectal variation in the outcomes of this experiment. This is consistent with Cuza et al. (2019).

[5]  In addition to representing the time at which children would have received at least half of their primary education in Spanish, students in this school temporarily received asynchronous instruction in third grade during the COVID-19 pandemic, such that their exposure to Spanish during this time would have been far less than under traditional circumstances. Therefore, only those children who had had consistent interaction with teachers and peers in Spanish prior to this time were included. This must be recognized as an unforeseeable yet considerable limitation of the present study.

[6]  In the indicative, third person singular –ar verbs such as those used in the present study end in the vowel /a/, while those with plural inflections end in /an/. Therefore, avoiding the third person raised the salience of DOM in this experiment by avoiding two /a/ vowels in succession.

[7] This task design is consistent with Montrul and Sánchez-Walker (2013) in which the children's experiment was kept brief due to younger participants' limited attentional resources, while adults completed lengthier tasks to measure the acquisition of additional structures and reduce recognizability of the target items.

[8] This measurement refers to the five contexts of language use targeted on the language questionnaire except for "at school". The sum of this and school language use formed the frequency of use score in the statistical modeling.

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
