# Peer review of "On the Acquisition of Differential Object Marking in Child Heritage Spanish: Bilingual Education, Exposure, and Age Effects (In Memory of Phoebe Search)"

_languages, doi:10.3390/languages9010026_

Round 1

Reviewer 1 Report

Comments and Suggestions for Authors

The article effectively shows the effect of instruction on language acquisition, which has important implications for parents and educators, and language policies. It also clearly shows age effects and it helps understand heritage language acquisition. 

I only had an issue with p. 18, l. 739-741 "...they allow those of us who are committed to promoting initiatives to maintain bilingualism to achieve social justice for bilingual speakers". My issue is that this is too important an idea to include it in a way that reads as an afterthought. It would require explaining what is meant by social justice, too. My suggestion is to develop it further, either here or in the conclusion, or to remove it. 

Comments on the Quality of English Language

Excellent paper, a pleasure to read. I only have a couple of edits:

p.1, l. 33 "it is essential understand". Insert "to"

p.17, l. 699 "proceeding", shouldn't it be "preceding"?

Author Response

Dear reviewers,

Please find my responses in the attached file.

Reviewer 2 Report

Comments and Suggestions for Authors

Author Response

(The authors gave the same response as above.)
